# HILBERT-GUIDED SPARSE LOCAL ATTENTION

**Yunge Li**
Department of Computer Science
Oakland University
Rochester Hill, MI, USA
yungeli@oakland.edu

**Lanyu Xu**
Department of Computer Science
Oakland University
Rochester Hill, MI, USA
lxu@oakland.edu

## ABSTRACT

The quadratic compute and memory costs of global self-attention severely limit its use in high-resolution images. Local attention reduces complexity by restricting attention to neighborhoods. Block-sparse kernels can further improve the efficiency of local attention, but conventional local attention patterns often fail to deliver significant speedups because tokens within a window are not contiguous in the 1D sequence. This work proposes a novel method for constructing windows and neighborhoods based on the Hilbert curve. Image tokens are first reordered along a Hilbert curve, and windows and neighborhoods are then formed on the reordered 1D sequence. From a block-sparse perspective, this strategy significantly increases block sparsity and can be combined with existing block-sparse kernels to improve the efficiency of 2D local attention. Experiments show that the proposed Hilbert Window Attention and Hilbert Slide Attention can accelerate window attention and slide attention by about $4\times$ and $18\times$, respectively. To assess practicality, the strategy is instantiated as the Hilbert Window Transformer and the Hilbert Neighborhood Transformer, both of which achieve end-to-end speedups with minimal accuracy loss. Overall, combining Hilbert-guided local attention with block-sparse kernels offers a general and practical approach to enhancing the efficiency of 2D local attention for images.

## 1 INTRODUCTION

In recent years, models based on self-attention mechanisms, especially the Transformer architecture, have achieved significant success in computer vision. However, the computational and memory requirements of global self-attention increase quadratically with sequence length, which severely limits its use in processing high-resolution images (Vaswani et al., 2017). To address this problem, local attention restricts each token's receptive field to its neighborhood and thereby reduces complexity.

Some typical works, such as Swin Transformer (Liu et al., 2021) and Neighborhood Attention Transformer (NAT) (Hassani et al., 2023) has shown that local attention can maintain the expressiveness of the model while considerably improving computational efficiency, making it a core focus in current research on efficient models. However, existing local attention methods still focus primarily on algorithm and structure design, while optimization at the kernel level remains limited. This is especially true for widely used 2D attention patterns in vision, such as window, sliding window, and neighborhood attention. Current kernel optimization techniques,

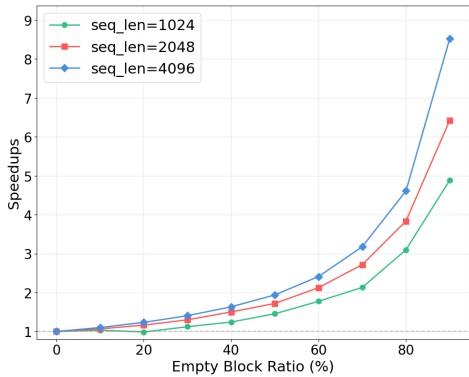

Figure 1: **Speedup from block sparsity.** When the sequence length is fixed, a higher empty blocks ratio leads to faster computation, with the effect especially pronounced at high sparsity ($> 80\%$). Additionally, longer sequences yield greater speedups.

The code is available at: https://github.com/Yunge6666/Hilbert-Local-Attention

for example, FlashAttention (Dao et al., 2022), are often designed for 1D sequences (e.g., text) or regularly structured sparse patterns and are not well adapted to 2D image local attention.

FlexAttention (Dong et al., 2024) introduces a more flexible approach to optimizing sparse attention, including local attention on 2D images. As a block-sparse attention framework, FlexAttention breaks down the attention operation into different types of blocks: *full blocks*, *partial blocks*, and *empty blocks*. Empty blocks are skipped in computation, which enables the efficient use of sparsity. Analysis of empty block ratio in FlexAttention (Figure 1) reveals that, for fixed sequence lengths such as 1024, 2048, and 4096, the speedup of attention computation is positively correlated with block sparsity, specifically the percentage of empty blocks. Therefore, a higher ratio of empty blocks reduces both computational and memory costs, leading to higher efficiency. This observation suggests that increasing the proportion of empty blocks is an effective way to accelerate attention computation.

However, the ratio of empty blocks is restricted in conventional local attention patterns. Window self-attention (WSA), sliding attention (SA), or neighborhood attention (NA) typically constructs windows/neighborhoods as regular squares in row-major order for 2D images. As shown in Figure 2, such row-major sequence misaligns with 2D neighborhoods, making tokens within a window discontinuous in the 1D sequence. This restricts the empty block ratio and produces many partial blocks, along with element-wise masking overhead (explained in Sec 3.1).

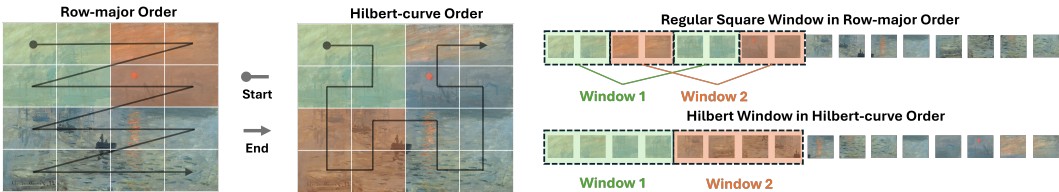

Figure 2: Row-major sequence vs. Hilbert-curve sequence

Motivated by these observations, this work proposes a novel method, Hilbert-guided construction of windows and neighborhoods, to enable more efficient use of sparsity by increasing the empty blocks ratio. Specifically, the Hilbert curve (Hilbert, 1935) is adopted due to its strong locality-preserving property (Jagadish, 1997): when mapping 2D image tokens to a 1D sequence, it maintains neighborhood relations. This reordering greatly increases the ratio of empty blocks in the local attention pattern and reduces the ratio of partial blocks. Within block-sparse kernel execution, empty blocks are skipped, directly reducing computational and memory overhead. Moreover, the programmable interface of the FlexAttention framework enables deployment without modifying the model or training pipeline and without hand-written kernels. In short, although local attention is inherently sparse, regular square windows in row-major order still produce many partial blocks that require computation. By remapping 2D local attention patterns into contiguous 1D blocks via the Hilbert reordering, the empty block ratio is greatly increased, leading to a significant acceleration of local attention.

The main contributions of this work are as follows: 1) A novel Hilbert-guided construction of local windows/neighborhoods is proposed, enabling highly sparse attention computation through Hilbert Window Attention (HWA), Hilbert Slide Attention (HSA), and Hilbert Neighborhood Attention (HNA); 2) These new attention patterns leverage block-sparse kernel to achieve significant efficiency gains; 3) HWA and HNA are instantiated into two end-to-end trainable models, Hilbert Window Transformer (HWT) and Hilbert Neighborhood Transformer (HNT), both of which achieve end-to-end speedups with minimal accuracy loss.

## 2 RELATED WORK

### 2.1 LOCAL ATTENTION FOR IMAGES

In Vision Transformer (Dosovitskiy et al., 2020), while globally computed full attention mechanisms are effective, they come with high computational costs. For high-dimensional data, such as images, their quadratic complexity becomes a significant bottleneck. Local attention mechanisms are based on a reasonable inductive bias: a visual token typically only needs to interact with tokens

in its surrounding neighborhood. This assumption significantly reduces computational complexity, making it a mainstream approach for building efficient vision transformer (Vaswani et al., 2021), (Yang et al., 2021), (Dong et al., 2022), (Tu et al., 2022),(Chen et al., 2021),(Liu et al., 2022), (Li et al., 2022).

Stand-Alone Self-Attention (SASA) (Ramachandran et al., 2019) was an early pioneering attempt to entirely replace convolutional layers with local self-attention layers in visual tasks. Swin Transformer (Liu et al., 2021) introduced a local attention mechanism based on regular windows and a shifted window strategy, cleverly enabling cross-window information exchange. It has become a milestone work in the field of vision transformers. The Slide Transformer (Pan et al., 2023) revisits sliding window attention by replacing the inefficient $Im2Col$ function with depthwise convolution and equipping it with a learned shift module, thereby achieving efficient local attention. NAT proposed neighborhood attention, which differs from previous sliding window attention in its boundary handling: instead of padding zeros, NAT repeats the same "window" at the boundaries.

Although local attention theoretically significantly reduces computational complexity, its practical efficiency highly depends on the quality of the underlying implementation. Many early implementations of local attention still required materializing large intermediate matrices in memory or handling complex masking logic, leading to significant memory overhead and low parallelization efficiency. As a result, they failed to realize the theoretical performance advantages fully. This bottleneck also underscores the need for fundamental optimization research at the kernel level.

## 2.2 ATTENTION KERNEL OPTIMIZATION

The goal of efficient attention computation techniques is to directly address the memory and computational bottlenecks of attention mechanisms at the system level. In recent years, breakthroughs in this field have primarily stemmed from the design of hardware-aware kernels (Zhang et al., 2024), (Kwon et al., 2023), (Liu et al., 2023), (Yuan et al., 2025), (Xu et al., 2025).

FlashAttention (Dao et al., 2022) is a foundational work. It was the first to restructure attention computation from the perspective of memory access cost, proposing an I/O-aware algorithm. Its core innovation lies in a tiling strategy that decomposes the computation to be performed in the GPU's high-speed SRAM, avoiding the materialization of large intermediate attention matrices in memory. This reduces the memory complexity and achieves a several-fold speedup. FlashAttentionv2 (Dao, 2023) further exploits hardware potential by restructuring parallelization strategies and refining task scheduling, significantly reducing kernel synchronization overhead and achieving higher throughput. The latest FlashAttentionv3 (Shah et al., 2024) begins to leverage features of newer-generation GPUs (e.g., NVIDIA H100) to further improve computational efficiency NATTEN also specifically conducted extreme kernel-level optimizations for Neighborhood Attention (Hassani et al., 2024), (Hassani et al., 2025). They eliminated the inherent $O(N^2)$ intermediate memory cost in neighborhood attention, providing a highly efficient implementation for this specific local attention pattern. XFormers (Lefaudeux et al., 2022) provides a library integrating multiple efficient attention implementations. XFormers includes not only memory-efficient attention implementations but also various predefined patterns.

However, the high-performance kernels of FlashAttention, NATTEN, and XFormers are all hand-optimized for predefined, fixed sparsity patterns. This introduces a key limitation: when research requires experimenting with a completely new, unimplemented attention pattern, significant effort is still needed to redesign and reimplement the underlying kernel, creating a high barrier to entry and a lack of flexibility. To address the flexibility issue that FlexAttention (Dong et al., 2024) was proposed to solve, a shift in programming paradigm is represented. FlexAttention builds a general compilation framework that allows developers to define arbitrary block-sparse attention patterns directly using high-level Python code. The compiler then automatically generates highly optimized GPU kernel code at compile time. This approach decouples algorithm innovation from low-level optimization. This work leverages the generality of FlexAttention to explore and further optimize efficiency bottlenecks of local attention in vision.

## 3  METHOD

### 3.1  HILBERT LOCAL ATTENTION

The proposed Hilbert-guided local attention is illustrated in Figure 3. Given $N$ tokens, attention is viewed as computation on an N×N matrix: rows correspond to each query ($q$) token, and columns correspond to all key ($k$) tokens. The attention pattern of 16 tokens is visualized into the $16 \times 16$ matrix.

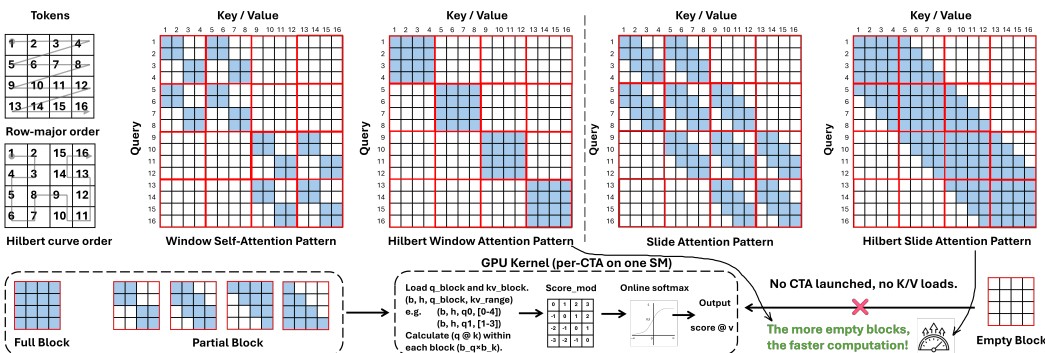

Figure 3: **Local attention patterns**. (1) In this case, the feature map is assumed to $4 \times 4$, therefore there are 16 tokens. The window size is $2 \times 2$, and $b_q = b_k = 4$. (2) Hilbert reordering not only preserves spatial locality but also increases the proportion of empty blocks in Hilbert local attention patterns, thereby reducing computational and memory access overhead. This further unleashes the potential of block-sparse attention and accelerates local attention computation.

Block-sparse attention (e.g., FlexAttention) tiles the $N \times N$ matrix into fixed-size blocks, each with a shape of $b_q \times b_k$ (outlined in red). Here $b_q = b_k = 4$. If all elements within a block participate in the computation, it is called a *full block*; if some elements are masked, it is a *partial block*; and if all elements in the block are masked, it is an *empty block*, which is skipped. Partial blocks cannot be skipped during computation and require element-wise masking, which introduces overhead and reduces efficiency, whereas dense computation in full blocks is generally faster. In terms of efficiency, empty blocks are preferable to full blocks, which are preferable to partial blocks. This mechanism transforms traditional dense self-attention into block-sparse attention based on the number of non-empty blocks.

In local attention, each query only attends to keys/values within a fixed window. Conventional local attention patterns typically partition the 2D space into regular square windows in row-major order. Given 16 tokens, with $2 \times 2$ window size, the first window takes (1,2,5,6) tokens to compute local attention, and the second takes (3,4,7,8) tokens. In the window attention pattern, attention (marked in blue) of these eight tokens forms four partial blocks, each of which is half full. For tokens not in the same window, the attention weight is masked as 0 (marked in white).

To increase the empty block ratio and reduce partial blocks, this paper introduces the Hilbert curve to replace row-major. By reordering the token sequence according to the Hilbert curve, windows or neighborhoods can be generated contiguously in the 1D sequence while preserving 2D spatial locality. With $2 \times 2$ window size, the first window takes (1,2,3,4) tokens to compute local attention, and the second takes (5,6,7,8) tokens. The tokens in each window are continuous in the 1D sequence, resulting in a more compact attention pattern. For the first eight tokens, HWA forms two full blocks and two empty blocks, increasing the empty block ratio from $0\%$ to $50\%$ and thereby accelerating attention computation. The same principle applies to the slide attention pattern. In this example, although the empty blocks ratio remains unchanged, the partial blocks ratio decreases (from $100\%$ to $50\%$), further improving efficiency. In general, HWA and HSA produce more empty blocks and fewer partial blocks than WSA and SA.

On GPUs, the overall runtime of block-sparse attention is jointly determined by the total workload and the effective parallel throughput (Blog, 2020). It can be approximated as follows:

$$T \approx \frac{\sum_{i=1}^{M} (\alpha + \beta \cdot r_i)}{P_{\text{eff}}} \tag{1}$$

where $\alpha$ represents the overhead per CTA (compute thread array), covering kernel/CTA launch, $q$ block loading, and initialization (Corporation, 2025a); $\beta$ denotes the unit cost of processing a single non-empty block, which includes loading $q$ block and $k/v$ block, computing $qk^\top$ within the block, score modification (e.g., relative position bias or Alibi (Press et al., 2021)), online softmax (Dao et al., 2022), and aggregation with value ($v$); $r_i$ is the number of non-empty blocks contained in the $i_{th}$ CTA; and $P_{\text{eff}}$ is the effective parallelism, determined by the number of streaming multiprocessors (SMs) and some memory factors (Corporation, 2025b). By skipping empty blocks, block-sparse attention avoids launching corresponding CTA and loading key/value data, thereby reducing computation and memory access. The proposed Hilbert reordering further accelerates block-sparse attention by increasing the ratio of empty blocks.

According to the Equation 1, more empty blocks reduce the value of $\sum_{i=1}^{M} (\alpha + \beta \cdot r_i)$, thereby shortening the total runtime. It should be emphasized that the combination of block size and window size directly affects the distribution of full, partial, and empty blocks, which in turn impacts overall performance. A reasonable block-window configuration can help increase the empty block ratio. A systematic evaluation is provided in Section 4.

## 3.2 HILBERT WINDOW TRANSFORMER

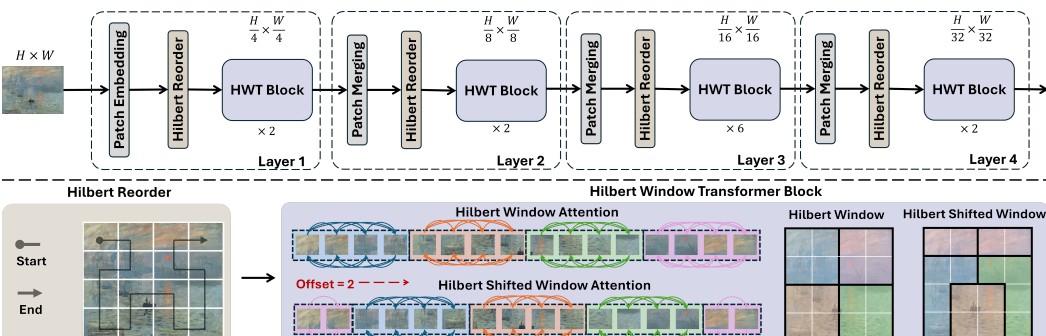

Figure 4: **The architecture of Hilbert Window Transformer.** After the token sequence is reordered according to the Hilbert curve, windows are constructed on the 1D sequence for window attention. Cross-window interaction is achieved by shifting these windows along the 1D sequence. Although the windows formed in the Hilbert-ordered sequence may correspond to irregular shapes in the original 2D space, the tokens within each window remain spatially adjacent in the 2D image.

WSA is applied in Swin Transformer, and it does not leverage block sparsity; instead, Swin Transformer performs dense WSA within each window. Although this approach implicitly partitions the sequence into windows, it does not further optimize the underlying computation. Block-sparse kernel can accelerate WSA; however, it may sometimes suffer from efficiency loss due to generating more partial blocks. By applying HWA, the number of empty blocks increases and the number of partial blocks decreases, thereby better leveraging the potential of block-sparse attention. Based on this idea, this work introduces the Hilbert Window Transformer (HWT). HWT shares a similar architecture with Swin Transformer, with the main differences lying in the window construction, the shift window operation, and the use of RPB. As shown in Figure 4, the image is first divided into tokens (or patches), which are then reordered according to the Hilbert curve path. Since feature maps with the same size produce the same Hilbert-curve path, a feature map of size $(H, W)$ corresponds to a unique path $P_{(H,W)}$. The path $P_{(H,W)}$ is computed and cached at model initialization, and whenever a feature map with the same size $(H, W)$ is processed, the cached path $P_{(H,W)}$ can be reused.

HWT blocks are also used in pairs: the first block performs HWA, and the second performs Hilbert Shifted Window Attention (HSWA). Thanks to the spatial locality-preserving property of the Hilbert

curve, Hilbert windows can be constructed directly on the 1D sequence, ensuring that tokens within each window remain adjacent in the original 2D space. Applying the shifted window strategy of Swin Transformer directly to HWT would make the attention mask overly complex, so HWT performs the window shift along the 1D sequence by moving each window forward by a fixed offset. This facilitates interaction between tokens from different windows and enhances the model's ability to capture global features. It should be noted that window shift may introduce tokens at the beginning and end of the sequence that are not adjacent in 2D space. Even if they fall into the same window, irrelevant attention connections must be masked out. In addition, since Hilbert windows and Hilbert shifted windows may exhibit irregular shapes, the window-based RPB used in Swin Transformer is no longer suitable. Instead, HWT enlarges the window to the full feature map, enabling a global relative position bias (global RPB).

Both HWA and HSWA can be implemented with the FlexAttention by configuring its $mask\_mod$ and the $score\_mod$ function, which expresses the block-sparse pattern induced by Hilbert reordering. As shown in Figure 3, this approach reduces memory access and computational overhead in invalid regions, better leverages block-sparse acceleration, and thus further improves the overall computational efficiency of the HWT model.

## 3.3 HILBERT NEIGHBORHOOD TRANSFORMER

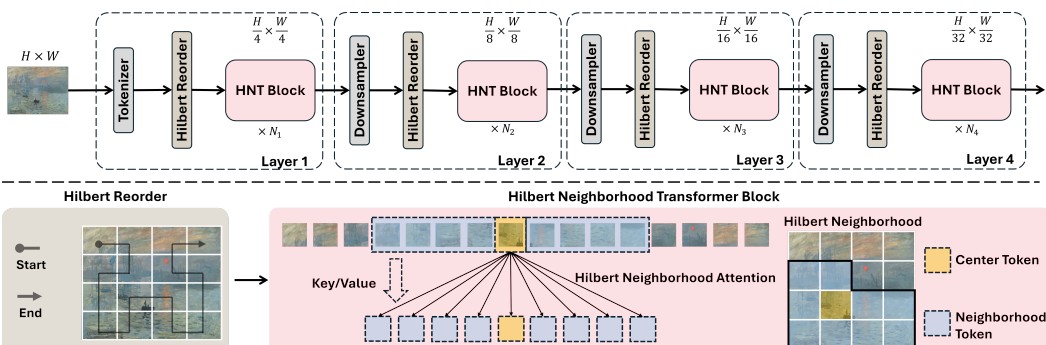

Figure 5: **The architecture of Hilbert Neighborhood Transformer.** Hilbert reordering enables the extraction of neighborhoods on the 1D token sequence while preserving spatial proximity. As a result, the 2D neighborhood attention can be converted into a 1D neighborhood attention.

Slide attention and neighborhood attention are mechanistically similar, differing primarily in their handling of boundary conditions. Both mechanisms suffer from high intermediate storage overhead, as each token must gather others within its neighborhood, typically requiring an $N^2$ tensor. This leads to memory and bandwidth constraints that impair training and inference efficiency, a key bottleneck observed in models like SASA and Slide Transformer. While methods such as FlashAttention and Xformer offer kernel-level optimizations for 1D slide attention, adapting these to images requires nontrivial kernel modifications due to their 2D nature. In contrast, NATTEN optimizes 2D neighborhood attention kernels by avoiding the explicit materialization of intermediates, and FlexAttention offers a flexible framework that supports both attention types. However, as shown in Figure 3, implementing SA/SASA with FlexAttention still produces many partial blocks, whereas HSA reduces the ratio of partial blocks and increases the number of empty blocks. Similarly, HNA attains a higher empty block ratio compared to NA.

Building on these advances, this work proposes the Hilbert Neighborhood Transformer (HNT), whose overall architecture is similar to the NAT, as shown in Figure 5. In HNT, after the image is converted into a token sequence, the tokens are reordered according to the Hilbert curve path. As with HWT, the Hilbert curve path can be cached and reused to reduce computational overhead. Similar to Slide Transformer and NAT, each token in HNT only attends to its surrounding tokens, a process carried out within the HNT block. Thanks to the ability of Hilbert reordering to preserve 2D spatial locality while mapping to a 1D sequence, each token only needs to attend to its neighboring tokens in the 1D sequence to effectively express 2D neighborhood relationships. Thus, the original 2D neighborhood attention is transformed into a 1D neighborhood attention. Specifically, 1D neighborhood attention can be implemented using the $na1d$ provided by NATTEN, or flexibly defined

using the $mask\_mod$ and $score\_mod$ functions in FlexAttention. These different implementations maintain consistency in attention patterns and computational logic, differing only in the underlying kernel optimization strategies. Therefore, they do not affect model accuracy but primarily differ in computational efficiency. Hilbert reordering increases the ratio of empty blocks while maintaining good spatial proximity, which further accelerates HNT.

## 4 EXPERIMENT

### 4.1 IMPLEMENTATION DETAILS

Model throughput and runtime depend on the hardware platform and software stack. Different GPU models, CUDA versions, or PyTorch versions can result in varying speedups. All experiments here are conducted on RTX 3080 GPU using CUDA 12.6 and PyTorch 2.7.0. For reference, the Appendix also reports results on an A100. The goal is to verify that Hilbert reordering increases the empty block ratio and thereby improves the efficiency of image local attention. Because the empty block ratio is mainly affected by input size, window/kernel size, and block size, we evaluate across different inputs, window sizes, and block sizes, comparing HWA with WSA and HSA/HNA with SA and NA. In addition, the implementation of WSA/SA/NA in FlexAttention is also evaluated and compared. We further include evaluations that combine input processing steps, such as window partitioning, Hilbert reordering, and QKV projection, with the attention computation. Finally, to assess feasibility in full models, we report the accuracy and efficiency of HWT and HNT on ImageNet (Deng et al., 2009). To show that HWA/HSA/HNA can be adapted to different optimized kernels without hand-written kernels, we further evaluate and compare them by applying FlashAttentionv2, xFormers, and NAT to them. Due to variability in CUDA timing, all evaluations are averages over 10 runs, each run consisting of 100 iterations with 25% warm-up and 75% measurement.

### 4.2 RESULTS AND ANALYSIS

Table 1: **Efficiency evaluation of window attention variants.** $I$ represents the input size, and $W$ represents the window size. All results in the table are obtained with batch size=16, head_num=2, head_dim=64, and block size=128. A complete evaluation is presented in the Appendix A.1.

| | Attention | Forward | | Backward | | Sparsity |
| --- | --- | --- | --- | --- | --- | --- |
| | | Time | Memory | Time | Memory | |
| | WSA | 0.28 ms | 32 MB | 1.06 ms | 104 MB | 0% |
| | WSA (xFormers) | 1.74 ms | 192 MB | 4.43 ms | 288 MB | – |
| $I = 64 \times 64, W = 8 \times 8$ | HWA (xFormers) | 0.24 ms | 17 MB | 1.12 ms | 163 MB | – |
| | WSA (Flex) | 0.40 ms | 17 MB | 1.79 ms | 65 MB | 87.50% |
| | HWA (Flex) | 0.12 ms (2.3×) | 17 MB | 0.69 ms | 65 MB | 96.88% |
| | WSA | 0.63 ms | 72 MB | 2.10 ms | 224 MB | 0% |
| | WSA (xFormers) | 3.98 ms | 432 MB | 9.96 ms | 648 MB | – |
| $I = 96 \times 96, W = 8 \times 8$ | HWA (xFormers) | 0.46 ms | 37 MB | 2.16 ms | 366 MB | – |
| | WSA (Flex) | 1.30 ms | 37 MB | 5.62 ms | 146 MB | 91.67% |
| | HWA (Flex) | 0.24 ms (2.6×) | 37 MB | 1.58 ms | 146 MB | 98.61% |
| | WSA | 1.03 ms | 164 MB | 2.85 ms | 408 MB | 0% |
| | WSA (xFormers) | 9.24 ms | 612 MB | 19.76 ms | 918 MB | – |
| $I = 96 \times 96, W = 12 \times 12$ | HWA (xFormers) | 0.78 ms | 37 MB | 3.41 ms | 284 MB | – |
| | WSA (Flex) | 2.02 ms | 37 MB | 8.23 ms | 146 MB | 87.50% |
| | HWA (Flex) | 0.59 ms (1.8×) | 37 MB | 2.87 ms | 146 MB | 96.14% |
| | WSA | 1.56 ms | 288 MB | 4.01 ms | 656 MB | 0% |
| | WSA (xFormers) | 12.17 ms | 864 MB | 33.22 ms | 1296 MB | – |
| $I = 96 \times 96, W = 16 \times 16$ | HWA (xFormers) | 0.44 ms | 37 MB | 2.40 ms | 256 MB | – |
| | WSA (Flex) | 2.63 ms | 37 MB | 10.47 ms | 146 MB | 83.33% |
| | HWA (Flex) | 0.40 ms (3.9×) | 37 MB | 2.16 ms | 146 MB | 97.22% |
| | WSA | 2.74 ms | 520 MB | 7.05 ms | 1160 MB | 0% |
| | WSA (xFormers) | 24.46 ms | 1536 MB | 60.79 ms | 2304 MB | – |
| $I = 128 \times 128, W = 16 \times 16$ | HWA (xFormers) | 0.79 ms | 66 MB | 4.74 ms | 455 MB | – |
| | WSA (Flex) | 5.68 ms | 66 MB | 22.60 ms | 260 MB | 87.50% |
| | HWA (Flex) | 0.68 ms (4.0×) | 66 MB | 3.87 ms | 260 MB | 98.44% |

**Window Attention Result.** Table 1 reports the efficiency of WSA and HWA with different optimized kernels under different input feature-map sizes and window sizes. WSA uses a dense kernel as the baseline. WSA (Flex) calls the block-sparse kernel in FlexAttention, but because it contains many partial blocks that require element-wise masking, it runs slower than WSA in all settings. The sparsity metric denotes the ratio of empty blocks, and HWA (Flex) shows higher sparsity than WSA (Flex) in every configuration. As a result, HWA (Flex) is faster than WSA (Flex) for both inference and training, and achieves up to $4.0\times$ speedup over the dense WSA. Benefiting from FlexAttention's integration of FlashAttentionv2, both WSA (Flex) and HWA (Flex) use much less memory than WSA in forward and backward. Memory consumption is dominated by the size of the QKV tensors and is only weakly affected by block size and window size. Thus, for the same input, WSA (Flex) and HWA (Flex) have the same memory usage. For example, when the input size is I=96, their memory stays unchanged across different window sizes. In contrast, WSA's memory grows with input and window size because dense computation materializes intermediate attention results within each window, whereas FlexAttention stores only lightweight block mask metadata and does not keep the full attention matrix. Overall, WSA (Flex) and HWA (Flex) reduce training and inference memory compared with WSA, and HWA (Flex) further improves computation speed. Since FlashAttentionv2 does not support arbitrary masks, WSA and HWA cannot be directly accelerated by it without manually modifying the kernel. Although xFormers supports more attention patterns, its implementation of WSA still uses an element-wise sparse path that stores many individual non-zero elements, which leads to high memory usage and is usually slower. In contrast, HWA can use block-diagonal fused dense kernels, so HWA (xFormers) in Table 1 is significantly faster than WSA (xFormers).

In Table 1, when the block size is fixed to 128 and the input size is the same (I=96), different window sizes change the sparsity and thus the runtime. Table 2 also reports how sparsity and speedups vary when the block size changes while input and window sizes are fixed. Therefore, for a given input size, an appropriate block-window configuration helps block-sparse local attention achieve better speedups. Appendix A.3 includes additional evaluations on the number of heads, head dimension, and batch size. These settings may produce different speedups on different hardware, but they do not affect block sparsity. Before comput-

Table 2: **Evaluation of different block sizes.** All results in the table are obtained with batch size=16, head_num=2, head_dim=64, input size=128 and window size =16.

| Attention | Block Size | Forward | | Sparsity |
| | | Time | Memory | |
|---|---|---|---|---|
| WSA | - | 2.74 ms | 520 MB | 0% |
| WSA (Flex) | 128 | 5.68 ms | 66 MB | 87.50% |
| | 256 | 5.68 ms | 66 MB | 87.50% |
| | 512 | 5.68 ms | 66 MB | 87.50% |
| | 1024 | 5.68 ms | 66 MB | 87.50% |
| HWA (Flex) | 128 | 0.68 ms (4.0x) | 66 MB | 98.44% |
| | 256 | 0.68 ms (4.0x) | 66 MB | 98.44% |
| | 512 | 1.43 ms (1.9x) | 66 MB | 96.88% |
| | 1024 | 2.78 ms (1.0x) | 66 MB | 93.75% |

ing attention, the input is processed with two steps: a reshape and linear projections to Q, K, and V. In WSA, the reshape is window partitioning. In HWA, the reshape is Hilbert reordering. WSA (Flex) does not require a reshape because FlexAttention operates directly on QKV obtained from the linear projection of the features. Figure 6a reports results of window attention variants for input $128 \times 128$ with a window $16 \times 16$. The attention time here differs from Table 1 because this evaluation includes RPB, but HWA (Flex) remains faster than WSA. In the QKV projection stage, HWA (Flex) is also faster than WSA. WSA first partitions windows, producing many small matrices and incurring more kernel launches and memory traffic, whereas HWA (Flex) processes fewer, larger matrices, which improves throughput. WSA (Flex) and HWA (Flex) both project the full input feature map, so their QKV projection times are essentially the same. Overall, HWA (Flex) takes 1.63 ms, which is $2.6\times$ faster than WSA's 4.22 ms and $3.9\times$ faster than WSA (Flex).

**Slide/Neighborhood Attention Results.** Table 3 reports results for SA and NA under different input sizes and kernel sizes. SA is an unoptimized baseline, and its runtime and memory usage are much higher than those of attention variants with an optimized kernel. SA can use the optimized kernels from xFormers and FlexAttention, while HSA can additionally be accelerated by FlashAttentionv2, because FlashAttentionv2 natively supports sliding-window (banded) patterns that match the HSA pattern. NA2D and HNA can be implemented with either FlexAttention or NATTEN. For SA(xFormers), it follows the same element-wise sparse path as WSA (xFormers) and is therefore

relatively slow, whereas HSA can directly use the optimized kernel in xFormers that is specifically designed for banded (sliding-window) attention, so it is much faster than SA (xFormers). Similarly, FlashAttentionv2 also provides a specialized optimized kernel for banded patterns, so HSA (FA2) achieves a very significant improvement in efficiency as well. Under the same settings, HSA (Flex) shows higher sparsity and larger speedups than SA (Flex), and HNA (Flex) shows higher sparsity and better speed than NA2D (Flex). With the NATTEN kernel, HNA (NAT) is also faster than NA2D. Concretely, at input $56 \times 56$ with kernel size $7 \times 7$, HNA (NAT) is $48.8\times$ faster than SA and $1.8\times$ faster than NA2D. At input $96 \times 96$ with kernel size $7 \times 7$, SA runs out of memory (OOM), and HSA (Flex) and HNA (NAT) are about $1.8$–$2.1\times$ faster than NA2D (NAT). Figure 6b

Table 3: **Efficiency evaluation of slide/neighborhood attention variants.** All results in the table are obtained with batch size=16, head_num=2, head_dim=64, and block size=128. A complete evaluation is presented in Appendix A.2.

| | Attention | Forward | | Backward | | Sparsity |
| --- | --- | --- | --- | --- | --- | --- |
| | | Time | Memory | Time | Memory | |
| $I = 56 \times 56, K = 7 \times 7$ | SA | 5.85 ms | 622 MB | 22.92 ms | 1835 MB | 0% |
| | HSA (FA2) | 0.25 ms | 13 MB | 0.75 ms | 76 MB | – |
| | SA (xFormers) | 1.31 ms | 141 MB | 3.56 ms | 208 MB | – |
| | HSA (xFormers) | 0.28 ms | 13 MB | 0.79 ms | 76 MB | – |
| | SA (Flex) | 0.56 ms | 13 MB | 1.89 ms | 50 MB | 80.17% |
| | HSA (Flex) | 0.32 ms | 13 MB | 1.32 ms | 50 MB | 87.84% |
| | NA2D (Flex) | 0.56 ms | 13 MB | 1.96 ms | 50 MB | 79.84% |
| | HNA (Flex) | 0.31 ms | 13 MB | 1.28 ms | 50 MB | 87.84% |
| | NA2D (NAT) | 0.21 ms | 13 MB | 2.65 ms | 75 MB | – |
| | HNA (NAT) | 0.12 ms | 13 MB | 1.03 ms | 75 MB | – |
| $I = 96 \times 96, K = 11 \times 11$ | SA | OOM | OOM | OOM | OOM | 0% |
| | HSA (FA2) | 0.69 ms | 37 MB | 2.08 ms | 218 MB | – |
| | SA (xFormers) | 6.58 ms | 545 MB | 18.71 ms | 818 MB | – |
| | HSA (xFormers) | 0.81 ms | 37 MB | 2.22 ms | 219 MB | – |
| | SA (Flex) | 1.83 ms | 37 MB | 7.82 ms | 146 MB | 87.89% |
| | HSA (Flex) | 0.61 ms | 37 MB | 3.23 ms | 146 MB | 95.87% |
| | NA2D (Flex) | 1.86 ms | 37 MB | 8.25 ms | 146 MB | 87.50% |
| | HNA (Flex) | 0.61 ms | 37 MB | 3.19 ms | 146 MB | 95.87% |
| | NA2D (NAT) | 1.07 ms | 37 MB | 8.38 ms | 219 MB | – |
| | HNA (NAT) | 0.51 ms | 37 MB | 3.06 ms | 219 MB | – |

presents an evaluation that combines attention with input processing. HSA and HNA require a re-

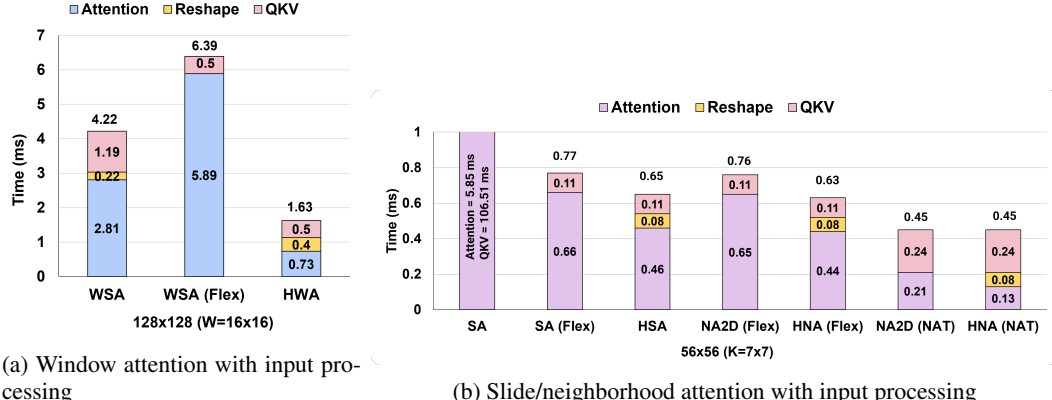

(a) Window attention with input processing

(b) Slide/neighborhood attention with input processing

Figure 6: **Comprehensive evaluation of combining attention computation with input processing.** (a) Combined overhead of different window attentions and their corresponding reshaping and QKV projection for a $128 \times 128$ input and a $16 \times 16$ window. (b) Combined overhead of different slide/neighborhood attentions and their corresponding reshaping and QKV projection for a $56 \times 56$ input and a $7 \times 7$ kernel size. Full results are provided in Appendix A.1 A.2.

Table 5: **Accuracy on CIFAR datasets.** Due to the low resolution of the CIFAR dataset, all models in the table are configured with only three layers, with depth=[2,6,4], head number=[3,6,12], and head dimention of 96. The patch size is 2, while the window size for SWIN and HWT is set to 4*4, and the kernel size for NAT is 7*7. The training epoch is 300 for all models.

| Model | Configuration | Window | CIFAR10 (32x32) | CIFAR100 (32x32) |
|---|---|---|---|---|
| SWIN | D=[2,6,4], H=[3,6,12] | 4x4 | 92.3% | 76.6% |
| HWT | D=[2,6,4], H=[3,6,12] | 16 | 92.2% | 76.4% |
| NAT | D=[2,6,4], H=[3,6,12] | 7x7 | 94.2% | 79.9% |
| HNT | D=[2,6,4], H=[3,6,12] | 49 | 94.1% | 79.9% |

shape step, specifically Hilbert reordering, whereas the other methods project the input directly to QKV and then compute attention. SA's neighborhood construction is both memory-intensive and time-consuming, making it non-competitive. Under the $I = 56 \times 56$ and $K = 7 \times 7$, NA2D (NAT) and HNA (NAT) have similar runtimes. At higher resolutions and larger kernel sizes, HNA (NAT) and HSA (Flex) are faster than the corresponding NA2D.

Table 4: Accuracy on ImageNet-1K.

Figure 7: Throughput comparison.

| | ImageNet-1K Top-1 Acc % | | | |
|---|---|---|---|---|
| | Swin-T | HWT-T | NAT-mini | HNT-mini |
| $224 \times 224$ | 81.2 | 81.0 | 81.8 | 81.6 |
| $256 \times 256$ | 81.6 | 81.5 | - | - |

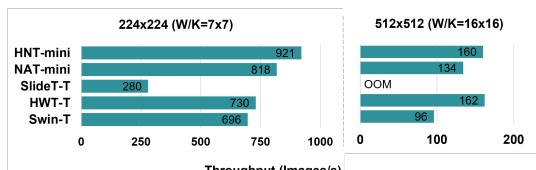

**HWT and HNT Results.** Training was conducted on 8 V100 GPUs for HWT-T and HNT-mini, with training settings and other hyperparameters matched to Swin-Tiny and NAT-mini. Because V100 does not support FlexAttention well, HWT used a dense kernel during training. Additional correctness tests verified that HWA with the dense kernel produces the same outputs as HWA with FlexAttention. Specifically, the correctness tests ensured that $mask\_mod$ and $score\_mod$ were designed so that the same attention pattern has identical outputs for the FlexAttention block-sparse kernel and for the dense implementation. Similar correctness checks were performed for HSA vs. SA and for HNA vs. NAT. Table 4 reports ImageNet-1K accuracy. At image resolutions of $224 \times 224$ and $256 \times 256$, applying the proposed HWA and HNA in HWT and HNT results in at most a $0.2\%$ drop in accuracy, whereas Figure 7 shows that both models deliver substantial efficiency gains, especially at high resolution. Table 5 shows the performance of HWT and HNT on CIFAR10 and CIFAR100. Compared with Swin and NAT, the accuracy of HWT and HNT is comparable, with performance differences within $0.2\%$. These results confirm the feasibility of the proposed HWA and HNA. Since HWT and HNT retain standard hierarchical backbone structures that are commonly used in object detection, semantic segmentation, and image or video generation pipelines, the same Hilbert local attention modules can, in principle, be reused in these settings as well. A comprehensive investigation into these downstream tasks will be conducted in future work.

## 5  CONCLUSION

This work proposes a Hilbert-guided construction of local windows and neighborhoods, from which HWA, HSA, and HNA are designed. Compared with conventional local attention, these variants increase block sparsity and, when combined with FlexAttention's programmable block-sparse kernel, significantly improve computational efficiency. Under the evaluation, HWA and HNA achieve several times inference speedups over WSA and SA, respectively. Building on this, the instantiated HWT and HNT deliver end-to-end speedups with minimal accuracy loss, confirming the feasibility and effectiveness of the approach. The combined *Hilbert-guided local attention + block-sparse kernel* approach is general and extensible, allowing it to interface with various block-sparse backends and model backbones. Overall, it provides a simple, practical path to system-level acceleration of 2D local attention for images and lays the groundwork for deployment across tasks and hardware.

ACKNOWLEDGMENT

This work was supported in part by the NSF under award #2245729.

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

# A APPENDIX

## A.1 FULL WINDOW ATTENTION RESULTS

Table 6: Full results of the efficiency evaluation of different Window Attention on RTX3080.

| | Attention | Forward | | Backward | | Sparsity |
|---|---|---|---|---|---|---|
| | | Time | Memory | Time | Memory | |
| $I = 56 \times 56, W = 7 \times 7$ | WSA | 0.22 ms | 22 MB | 0.73 ms | 70 MB | 0% |
| | WSA (xFormers) | 1.21 ms | 141 MB | 3.15 ms | 209 MB | - |
| | HWA (xFormers) | 0.26 ms | 13 MB | 1.05 ms | 140 MB | - |
| | WSA (Flex) | 0.59 ms | 13 MB | 1.95 ms | 50 MB | 79.84% |
| | HWA (Flex) | 0.31 ms (0.7x) | 13 MB | 1.20 ms | 50 MB | 87.84% |
| $I = 64 \times 64, W = 8 \times 8$ | WSA | 0.28 ms | 32 MB | 1.06 ms | 104 MB | 0% |
| | WSA (xFormers) | 1.74 ms | 192 MB | 4.43 ms | 288 MB | - |
| | HWA (xFormers) | 0.24 ms | 17 MB | 1.12 ms | 163 MB | - |
| | WSA (Flex) | 0.40 ms | 17 MB | 1.79 ms | 65 MB | 87.50% |
| | HWA (Flex) | 0.12 ms (2.3x) | 17 MB | 0.69 ms | 65 MB | 96.88% |
| $I = 96 \times 96, W = 8 \times 8$ | WSA | 0.63 ms | 72 MB | 2.10 ms | 224 MB | 0% |
| | WSA (xFormers) | 3.98 ms | 432 MB | 9.96 ms | 648 MB | - |
| | HWA (xFormers) | 0.46 ms | 37 MB | 2.16 ms | 366 MB | - |
| | WSA (Flex) | 1.30 ms | 37 MB | 5.62 ms | 146 MB | 91.67% |
| | HWA (Flex) | 0.24 ms (2.6x) | 37 MB | 1.58 ms | 146 MB | 98.61% |
| $I = 96 \times 96, W = 12 \times 12$ | WSA | 1.03 ms | 164 MB | 2.85 ms | 408 MB | 0% |
| | WSA (xFormers) | 9.24 ms | 612 MB | 19.76 ms | 918 MB | - |
| | HWA (xFormers) | 0.78 ms | 37 MB | 3.41 ms | 284 MB | - |
| | WSA (Flex) | 2.02 ms | 37 MB | 8.23 ms | 146 MB | 87.50% |
| | HWA (Flex) | 0.59 ms (1.8x) | 37 MB | 2.87 ms | 146 MB | 96.14% |
| $I = 96 \times 96, W = 16 \times 16$ | WSA | 1.56 ms | 288 MB | 4.01 ms | 656 MB | 0% |
| | WSA (xFormers) | 12.17 ms | 864 MB | 33.22 ms | 1296 MB | - |
| | HWA (xFormers) | 0.44 ms | 37 MB | 2.40 ms | 256 MB | - |
| | WSA (Flex) | 2.63 ms | 37 MB | 10.47 ms | 146 MB | 83.33% |
| | HWA (Flex) | 0.40 ms (3.9x) | 37 MB | 2.16 ms | 146 MB | 97.22% |
| $I = 128 \times 128, W = 16 \times 16$ | WSA | 2.74 ms | 520 MB | 7.05 ms | 1160 MB | 0% |
| | WSA (xFormers) | 24.46 ms | 1536 MB | 60.79 ms | 2304 MB | - |
| | HWA (xFormers) | 0.79 ms | 66 MB | 4.74 ms | 455 MB | - |
| | WSA (Flex) | 5.68 ms | 66 MB | 22.60 ms | 260 MB | 87.50% |
| | HWA (Flex) | 0.68 ms (4.0x) | 66MB | 3.87 ms | 260 MB | 98.44% |
| $I = 128 \times 256, W = 16 \times 16$ | WSA | 5.46 ms | 1024 MB | 14.02 ms | 2312 MB | 0% |
| | WSA (xFormers) | OOM | OOM | OOM | OOM | - |
| | HWA (xFormers) | 1.52 ms | 132 MB | 8.03 ms | 910 MB | - |
| | WSA (Flex) | 11.79 ms | 132 MB | 43.32 ms | 520 MB | 93.75% |
| | HWA (Flex) | 1.36 ms (4.0x) | 132 MB | 7.71ms | 520 MB | 99.22% |
| $I = 160 \times 160, W = 20 \times 20$ | WSA | 6.75 ms | 1252 MB | 15.25 ms | 2712 MB | 0% |
| | WSA (xFormers) | 55.16 ms | 3300 MB | 150.68 ms | 4950 MB | - |
| | HWA (xFormers) | 2.65 ms | 103 MB | 10.50 ms | 674 MB | - |
| | WSA (Flex) | 15.28 ms | 103 MB | 57.32 ms | 406 MB | 87.50% |
| | HWA (Flex) | 2.63 ms (2.6x) | 103 MB | 12.15 ms | 406 MB | 97.58% |

At low resolutions such as 56×56 with a window size of 7×7, HWA (Flex) fails to outperform WSA, primarily due to the limited sparsity under this configuration, which is insufficient to yield meaningful acceleration. This observation indicates that surpassing the performance of a kernel operating solely on full blocks requires a certain degree of sparsity—that is, a sufficiently high proportion of empty blocks must be skipped. Moreover, even at high resolutions, achieving ideal acceleration still depends on selecting appropriate window and block sizes.

Table 7 reports results on an A100. The A100 setup also uses CUDA 12.6 and PyTorch 2.7.0, and the test parameters are identical to those on the RTX 3080. Compared with the 3080 results, memory usage and sparsity remain unchanged because the attention pattern does not change. Runtimes differ and are faster on the A100, as expected. The A100 also supports higher resolutions such as $256 \times 256$, where HWA (Flex) shows a more pronounced speedup.

Table 7: Full results of the efficiency evaluation of different Window Attention on A100. All results in the table are obtained with batch size=16, head_num=2, head_dim=64, and block size=128.

| | Attention | Forward | | Backward | | Sparsity |
| | | Time | Memory | Time | Memory | |
|---|---|---|---|---|---|---|
| $I = 56 \times 56, W = 7 \times 7$ | WSA | 0.16 ms | 22 MB | 0.44 ms | 70 MB | 0% |
| | WSA (Flex) | 0.30 ms | 13 MB | 1.03 ms | 50 MB | 79.84% |
| | HWA (Flex) | 0.18 ms | 13 MB | 0.66 ms | 50 MB | 87.84% |
| $I = 64 \times 64, W = 8 \times 8$ | WSA | 0.17 ms | 32 MB | 0.57 ms | 104 MB | 0% |
| | WSA (Flex) | 0.20 ms | 17 MB | 0.80 ms | 65 MB | 87.50% |
| | HWA (Flex) | 0.19 ms | 17 MB | 0.39 ms | 65 MB | 96.88% |
| $I = 96 \times 96, W = 8 \times 8$ | WSA | 0.35 ms | 72 MB | 1.21 ms | 224 MB | 0% |
| | WSA (Flex) | 0.63 ms | 37 MB | 2.39 ms | 146 MB | 91.67% |
| | HWA (Flex) | 0.19 ms | 37 MB | 0.84 ms | 146 MB | 98.61% |
| $I = 96 \times 96, W = 12 \times 12$ | WSA | 0.77 ms | 164 MB | 1.69 ms | 408 MB | 0% |
| | WSA (Flex) | 0.98 ms | 37 MB | 3.45 ms | 146 MB | 87.50% |
| | HWA (Flex) | 0.33 ms | 37 MB | 1.24 ms | 146 MB | 96.14% |
| $I = 96 \times 96, W = 16 \times 16$ | WSA | 0.90 ms | 288 MB | 2.21 ms | 656 MB | 0% |
| | WSA (Flex) | 1.29 ms | 37 MB | 4.31 ms | 146 MB | 83.33% |
| | HWA (Flex) | 0.21 ms | 37 MB | 0.97 ms | 146 MB | 97.22% |
| $I = 128 \times 128, W = 16 \times 16$ | WSA | 1.55 ms | 520 MB | 3.85 ms | 1160 MB | 0% |
| | WSA (Flex) | 2.45 ms | 66 MB | 9.30 ms | 260 MB | 87.50% |
| | HWA (Flex) | 0.35 ms | 66MB | 1.65 ms | 260 MB | 98.44% |
| $I = 256 \times 256, W = 32 \times 32$ | WSA | 21.06 ms | 1024 MB | 44.19 ms | 2312 MB | 0% |
| | WSA (Flex) | 23.48 ms | 132 MB | 70.30 ms | 520 MB | 93.75% |
| | HWA (Flex) | 3.44 ms | 132 MB | 16.73 ms | 520 MB | 99.22% |
| $I = 160 \times 160, W = 20 \times 20$ | WSA | 3.71 ms | 1252 MB | 8.20 ms | 2712 MB | 0% |
| | WSA (Flex) | 9.17 ms | 103 MB | 24.10 ms | 406 MB | 87.50% |
| | HWA (Flex) | 1.33 ms | 103 MB | 4.87 ms | 406 MB | 97.58% |

Table 8: Full results of the comprehensive evaluation of Window Attention combined with additional input processing.

| | Attention | RTX3080 | | | | A100 | | | |
| | | Attention | Reshape | QKV | Foward | Attention | Reshape | QKV | Foward |
|---|---|---|---|---|---|---|---|---|---|
| I=56x56 W=7x7 | WSA | 0.23 ms | 0.05 ms | 0.24 ms | 0.52 ms | 0.16 ms | 0.05 ms | 0.17 ms | 0.38 ms |
| | WSA (Flex) | 0.72 ms | 0 ms | 0.11 ms | 0.83 ms | 0.33 ms | 0 ms | 0.08 ms | 0.41 ms |
| | HWA (Flex) | 0.44 ms | 0.08 ms | 0.11 ms | 0.63 ms | 0.22 ms | 0.09 ms | 0.08 ms | 0.39 ms |
| I=64x64 W=8x8 | WSA | 0.29 ms | 0.06 ms | 0.31 ms | 0.66 ms | 0.18 ms | 0.06 ms | 0.21 ms | 0.45 ms |
| | WSA (Flex) | 0.41 ms | 0 ms | 0.14 ms | 0.55 ms | 0.26 ms | 0 ms | 0.09 ms | 0.35 ms |
| | HWA (Flex) | 0.12 ms | 0.10 ms | 0.14 ms | 0.36 ms | 0.21 ms | 0.11 ms | 0.09 ms | 0.41 ms |
| I=96x96 W=8x8 | WSA | 0.66 ms | 0.13 ms | 0.65 ms | 1.44 ms | 0.38 ms | 0.11 ms | 0.45 ms | 0.94 ms |
| | WSA (Flex) | 1.39 ms | 0 ms | 0.29 ms | 1.68 ms | 0.79 ms | 0 ms | 0.19 ms | 0.98 ms |
| | HWA (Flex) | 0.25 ms | 0.22 ms | 0.29 ms | 0.76 ms | 0.21 ms | 0.24 ms | 0.19 ms | 0.64 ms |
| I=96x96 W=12x12 | WSA | 1.11 ms | 0.13 ms | 0.65 ms | 1.89 ms | 0.82 ms | 0.11 ms | 0.47 ms | 1.40 ms |
| | WSA (Flex) | 2.07 ms | 0 ms | 0.29 ms | 2.36 ms | 1.14 ms | 0 ms | 0.19 ms | 1.33 ms |
| | HWA (Flex) | 0.61 ms | 0.22 ms | 0.29 ms | 1.12 ms | 0.28 ms | 0.24 ms | 0.19 ms | 0.71 ms |
| I=96x96 W=16x16 | WSA | 1.59 ms | 0.13 ms | 0.65 ms | 2.37 ms | 1.04 ms | 0.11 ms | 0.53 ms | 1.68 ms |
| | WSA (Flex) | 2.72 ms | 0 ms | 0.29 ms | 3.01 ms | 1.44 ms | 0 ms | 0.19 ms | 1.63 ms |
| | HWA (Flex) | 0.41 ms | 0.22 ms | 0.29 ms | 0.92 ms | 0.22 ms | 0.24 ms | 0.19 ms | 0.65 ms |
| I=128x128 W=16x16 | WSA | 2.81 ms | 0.22 ms | 1.19 ms | 4.22 ms | 1.81 ms | 0.20 ms | 0.92 ms | 2.93 ms |
| | WSA (Flex) | 5.89 ms | 0 ms | 0.50 ms | 6.39 ms | 3.47 ms | 0 ms | 0.32 ms | 3.79 ms |
| | HWA (Flex) | 0.73 ms | 0.4 ms | 0.50 ms | 1.63 ms | 0.30 ms | 0.42 ms | 0.32 ms | 1.04 ms |
| I=160x160 W=20x20 | WSA | 6.92 ms | 0.34 ms | 1.85 ms | 9.11 ms | 4.54 ms | 0.30 ms | 1.18 ms | 6.02 ms |
| | WSA (Flex) | 16.09 ms | 0 ms | 0.78 ms | 16.87 ms | 10.7 ms | 0 ms | 0.39 ms | 11.09 ms |
| | HWA (Flex) | 2.72 ms | 0.62 ms | 0.78 ms | 4.12 ms | 1.14 ms | 0.62 ms | 0.39 ms | 2.15 ms |

Table 8 presents end-to-end results on RTX 3080 and A100 for window-attention variants, combining the corresponding reshape and QKV projection under different input and window configurations. At $56 \times 56$ resolution, HWA (Flex) does not achieve good speedups, mainly because the sparsity

is low and the attention part is not significantly accelerated. As the resolution increases and the window size is chosen appropriately, the speedup becomes substantial, especially on A100, where at I=128, W=16, the acceleration approaches $3\times$.

## A.2 FULL SLIDE/NEIGHBORHOOD ATTENTION RESULTS

Table 9: Full results of the efficiency evaluation of different Slide/Neighborhood Attention on A100. All results in the table are obtained with batch size=16, head_num=2, head_dim=64, and block size=128.

| | Attention | Forward | | Backward | | Sparsity |
|---|---|---|---|---|---|---|
| | | Time | Memory | Time | Memory | |
| $I = 56 \times 56, W = 7 \times 7$ | SA | 3.84 ms | 622 MB | 16.06 ms | 1835 MB | 0% |
| | SA (Flex) | 0.28 ms | 13 MB | 0.98 ms | 50 MB | 80.17% |
| | HSA (Flex) | 0.21 ms | 13 MB | 0.74 ms | 50 MB | 87.84% |
| | NA2D (Flex) | 0.28 ms | 13 MB | 0.97 ms | 50 MB | 79.84% |
| | HNA (Flex) | 0.21 ms | 13 MB | 0.73 ms | 50 MB | 87.84% |
| $I = 64 \times 64, W = 9 \times 9$ | SA | 8.26 ms | 1332 MB | 34.05 ms | 3940 MB | 0% |
| | SA (Flex) | 0.34 ms | 17 MB | 0.96 ms | 65 MB | 84.96% |
| | HSA (Flex) | 0.20 ms | 17 MB | 0.60 ms | 65 MB | 90.82% |
| | NA2D (Flex) | 0.34 ms | 17 MB | 0.99 ms | 65 MB | 84.38% |
| | HNA (Flex) | 0.20 ms | 17 MB | 0.59 ms | 65 MB | 90.82% |
| $I = 96 \times 96, W = 9 \times 9$ | SA | 18.57 ms | 3070 MB | 84.14 ms | 8938 MB | 0% |
| | SA (Flex) | 0.98 ms | 37 MB | 3.05 ms | 146 MB | 88.73% |
| | HSA (Flex) | 0.35 ms | 37 MB | 1.38 ms | 146 MB | 95.87% |
| | NA2D (Flex) | 0.99 ms | 37 MB | 3.43 ms | 146 MB | 88.58% |
| | HNA (Flex) | 0.35 ms | 37 MB | 1.42 ms | 146 MB | 95.87% |
| $I = 96 \times 96, W = 11 \times 11$ | SA | OOM | OOM | OOM | OOM | 0% |
| | SA (Flex) | 1.04 ms | 37 MB | 3.83 ms | 146 MB | 87.89% |
| | HSA (Flex) | 0.35 ms | 37 MB | 1.39 ms | 146 MB | 95.87% |
| | NA2D (Flex) | 1.04 ms | 37 MB | 3.50 ms | 146 MB | 87.50% |
| | HNA (Flex) | 0.35 ms | 37 MB | 1.41 ms | 146 MB | 95.87% |
| $I = 96 \times 96, W = 17 \times 17$ | SA | OOM | OOM | OOM | OOM | 0% |
| | SA (Flex) | 1.62 ms | 37 MB | 4.76 ms | 146 MB | 81.06% |
| | HSA (Flex) | 0.52 ms | 37 MB | 2.22 ms | 146 MB | 93.17% |
| | NA2D (Flex) | 1.66 ms | 37 MB | 5.57 ms | 146 MB | 80.40% |
| | HNA (Flex) | 0.52 ms | 37 MB | 2.20 ms | 146 MB | 93.17% |
| $I = 128 \times 128, W = 17 \times 17$ | SA | OOM | OOM | OOM | OOM | 0% |
| | SA (Flex) | 3.82 ms | 66 MB | 9.86 ms | 260 MB | 87.16% |
| | HSA (Flex) | 0.89 ms | 66 MB | 3.20 ms | 260 MB | 96.13% |
| | NA2D (Flex) | 3.88 ms | 66 MB | 10.27 ms | 260 MB | 86.72% |
| | HNA (Flex) | 0.88 ms | 66 MB | 3.17 ms | 260 MB | 96.13% |

Table 9 reports all results for the Slide/Neighborhood Attention variants on A100. The same library versions and parameter settings used on the RTX 3080 are employed. The A100 can handle computations at $96 \times 96$ resolution, but OOM occurs when the kernel size is further increased. Even so, HSA (Flex) and HNA (Flex) still show strong speedups. Due to hardware resource constraints, the NATTEN library could not be installed on the A100 machine, so NAT-related attention variants were not evaluated.

Table 10: Full results of the comprehensive evaluation of Slide/Neighborhood Attention combined with additional input processing.

| | Attention | RTX3080 | | | | A100 | | | |
|---|---|---|---|---|---|---|---|---|---|
| | | Attention | Reshape | QKV | Foward | Attention | Reshape | QKV | Foward |
| I=56x56 K=7x7 | SA | 5.85 ms | 0 ms | 106.51 ms | 112.36 ms | 3.84 ms | 0 ms | 77.2 ms | 81.04 ms |
| | SA (Flex) | 0.66 ms | 0 ms | 0.11 ms | 0.77 ms | 0.35 ms | 0 ms | 0.08 ms | 0.43 ms |
| | HSA (Flex) | 0.46 ms | 0.08 ms | 0.11 ms | 0.65 ms | 0.22 ms | 0.09 ms | 0.08 ms | 0.39 ms |
| | NA2D (Flex) | 0.65 ms | 0 ms | 0.11 ms | 0.77 ms | 0.34 ms | 0 ms | 0.08 ms | 0.42 ms |
| | HNA (Flex) | 0.44 ms | 0.08 ms | 0.11 ms | 0.63 ms | 0.22 ms | 0.09 ms | 0.08 ms | 0.39 ms |
| | NA2D (NAT) | 0.21 ms | 0 ms | 0.24 ms | 0.45 ms | - | - | - | - |
| | HNA (NAT) | 0.13 ms | 0.08 ms | 0.24 ms | 0.45 ms | - | - | - | - |
| I=64x64 K=9x9 | SA | 12.65 ms | 0 ms | 446.85 ms | 459.5 ms | 8.26 ms | 0 ms | 307.51 ms | 315.77 ms |
| | SA (Flex) | 0.52 ms | 0 ms | 0.14 ms | 0.66 ms | 0.38 ms | 0 ms | 0.09 ms | 0.47 ms |
| | HSA (Flex) | 0.29 ms | 0.1 ms | 0.14 ms | 0.53 ms | 0.21 ms | 0.24 ms | 0.19 ms | 0.64 ms |
| | NA2D (Flex) | 0.51 ms | 0 ms | 0.14 ms | 0.65 ms | 0.38 ms | 0 ms | 0.09 ms | 0.47 ms |
| | HNA (Flex) | 0.29 ms | 0.1 ms | 0.14 ms | 0.53 ms | 0.21 ms | 0.24 ms | 0.19 ms | 0.64 ms |
| | NA2D (NAT) | 0.28 ms | 0 ms | 0.30 ms | 0.58 ms | - | - | - | - |
| | HNA (NAT) | 0.20 ms | 0.1 ms | 0.30 ms | 0.60 ms | - | - | - | - |
| I=96x96 K=9x9 | SA | OOM | OOM | OOM | OOM | 18.57 ms | 0 ms | 647.88 ms | 666.45 ms |
| | SA (Flex) | 1.78 ms | 0 ms | 0.29 ms | 2.07 ms | 1.1 ms | 0 ms | 0.19 ms | 1.29 ms |
| | HSA (Flex) | 0.63 ms | 0.22 ms | 0.29 ms | 1.14 ms | 0.31 ms | 0.24 ms | 0.19 ms | 0.74 ms |
| | NA2D (Flex) | 1.79 ms | 0 ms | 0.29 ms | 2.08 ms | 1.11 ms | 0 ms | 0.19 ms | 1.30 ms |
| | HNA (Flex) | 0.64 ms | 0.22 ms | 0.29 ms | 1.15 ms | 0.31 ms | 0.24 ms | 0.19 ms | 0.74 ms |
| | NA2D (NAT) | 0.59 ms | 0 ms | 0.65 ms | 1.24 ms | - | - | - | - |
| | HNA (NAT) | 0.44 ms | 0.22 ms | 0.65 ms | 1.31 ms | - | - | - | - |
| I=96x96 K=11x11 | SA | OOM | OOM | OOM | OOM | OOM | OOM | OOM | OOM |
| | SA (Flex) | 1.90 ms | 0 ms | 0.29 ms | 2.19 ms | 1.18 ms | 0 ms | 0.19 ms | 1.37 ms |
| | HSA (Flex) | 0.63 ms | 0.22 ms | 0.29 ms | 1.14 ms | 0.31 ms | 0.24 ms | 0.19 ms | 0.74 ms |
| | NA2D (Flex) | 1.95 ms | 0 ms | 0.29 ms | 2.24 ms | 0.31 ms | 0 ms | 0.19 ms | 0.50 ms |
| | HNA (Flex) | 0.64 ms | 0.22 ms | 0.29 ms | 1.15 ms | 0.64 ms | 0.24 ms | 0.19 ms | 1.07 ms |
| | NA2D (NAT) | 1.07 ms | 0 ms | 0.65 ms | 1.72 ms | - | - | - | - |
| | HNA (NAT) | 0.51 ms | 0.22 ms | 0.65 ms | 1.38 ms | - | - | - | - |
| I=96x96 K=17x17 | SA | OOM | OOM | OOM | OOM | OOM | OOM | OOM | OOM |
| | SA (Flex) | 2.93 ms | 0 ms | 0.29 ms | 3.22 ms | 1.80 ms | 0 ms | 0.19 ms | 1.99 ms |
| | HSA (Flex) | 1.01 ms | 0.22 ms | 0.29 ms | 1.52 ms | 0.57 ms | 0.24 ms | 0.19 ms | 1.00 ms |
| | NA2D (Flex) | 3.00 ms | 0 ms | 0.29 ms | 3.29 ms | 1.81 ms | 0 ms | 0.19 ms | 2.00 ms |
| | HNA (Flex) | 1.03 ms | 0.22 ms | 0.29 ms | 1.54 ms | 0.57 ms | 0.24 ms | 0.19 ms | 1.00 ms |
| | NA2D (NAT) | 1.27 ms | 0 ms | 0.65 ms | 1.92 ms | - | - | - | - |
| | HNA (NAT) | 0.77 ms | 0.22 ms | 0.65 ms | 1.64 ms | - | - | - | - |
| I=128x128 K=17x17 | SA | OOM | OOM | OOM | OOM | OOM | OOM | OOM | OOM |
| | SA (Flex) | 6.47 ms | 0 ms | 0.50 ms | 6.97 ms | 4.10 ms | 0 ms | 0.32 ms | 4.42 ms |
| | HSA (Flex) | 1.78 ms | 0.40 ms | 0.50 ms | 2.68 ms | 0.94 ms | 0.42 ms | 0.32 ms | 1.68 ms |
| | NA2D (Flex) | 6.71 ms | 0 ms | 0.50 ms | 7.21 ms | 4.10 ms | 0 ms | 0.32 ms | 4.42 ms |
| | HNA (Flex) | 1.80 ms | 0.40 ms | 0.50 ms | 2.70 ms | 0.94 ms | 0.42 ms | 0.32 ms | 1.68 ms |
| | NA2D (NAT) | 2.25 ms | 0 ms | 1.14 ms | 3.39 ms | - | - | - | - |
| | HNA (NAT) | 1.37 ms | 0.40 ms | 1.14 ms | 2.91 ms | - | - | - | - |

Table 10 presents the full results for RTX 3080 and A100, including slide/neighborhood attention variants, which combine the corresponding reshape and QKV projection under various input and window configurations. At higher resolutions and larger kernel sizes, HNA and HSA perform better than the other attentions.

Table 11 reports all results for the Slide/Neighborhood Attention variants on RTX 3080. Due to memory and bandwidth limits, SA runs out of memory once the resolution reaches $96 \times 96$. At higher resolutions such as $128 \times 128$, HNA and HSA are more than $3\times$ faster than SA (Flex) and NA2D (Flex), and they are also faster than NA2D (NAT).

Table 11: Full results of the efficiency evaluation of different Slide/Neighborhood Attention on RTX 3080.

| | Attention | Forward | | Backward | | Sparsity |
|---|---|---|---|---|---|---|
| | | Time | Memory | Time | Memory | |
| $I = 56 \times 56, W = 7 \times 7$ | SA | 5.85 ms | 622 MB | 22.92 ms | 1835 MB | 0% |
| | HSA (FA2) | 0.25 ms | 13 MB | 0.75 ms | 76 MB | - |
| | SA (xFormers) | 1.31 ms | 141 MB | 3.56 ms | 208 MB | - |
| | HSA (xFormers) | 0.28 ms | 13 MB | 0.79 ms | 76 MB | - |
| | SA (Flex) | 0.56 ms | 13 MB | 1.89 ms | 50 MB | 80.17% |
| | HSA (Flex) | 0.32 ms | 13 MB | 1.32 ms | 50 MB | 87.84% |
| | NA2D (Flex) | 0.56 ms | 13 MB | 1.96 ms | 50 MB | 79.84% |
| | HNA (Flex) | 0.31 ms | 13 MB | 1.28 ms | 50 MB | 87.84% |
| | NA2D (NAT) | 0.21 ms | 13 MB | 2.65 ms | 75 MB | - |
| | HNA (NAT) | 0.12 ms | 13 MB | 1.03 ms | 75 MB | - |
| $I = 64 \times 64, W = 9 \times 9$ | SA | 12.65 ms | 1332 MB | OOM | OOM | 0% |
| | HSA (FA2) | 0.31 ms | 17 MB | 0.95 ms | 97 MB | - |
| | SA (xFormers) | 2.47 ms | 204 MB | 6.00 ms | 306 MB | - |
| | HSA (xFormers) | 0.37 ms | 17 MB | 1.02 ms | 98 MB | - |
| | SA (Flex) | 0.49 ms | 17 MB | 2.21 ms | 65 MB | 84.96% |
| | HSA (Flex) | 0.28 ms | 17 MB | 1.39 ms | 65 MB | 90.82% |
| | NA2D (Flex) | 0.49 ms | 17 MB | 2.16 ms | 65 MB | 84.38% |
| | HNA (Flex) | 0.28ms | 17 MB | 1.39 ms | 65 MB | 90.82% |
| | NA2D (NAT) | 0.28 ms | 17 MB | 3.52 ms | 98 MB | - |
| | HNA (NAT) | 0.20 ms | 17 MB | 1.37 ms | 98 MB | - |
| $I = 96 \times 96, W = 9 \times 9$ | SA | OOM | OOM | OOM | OOM | 0% |
| | HSA (FA2) | 0.69 ms | 37 MB | 2.08 ms | 218 MB | - |
| | SA (xFormers) | 4.79 ms | 462 MB | 13.49 ms | 693 MB | - |
| | HSA (xFormers) | 0.79 ms | 37 MB | 2.22 ms | 219 MB | - |
| | SA (Flex) | 1.68 ms | 37 MB | 7.32 ms | 146 MB | 88.73% |
| | HSA (Flex) | 0.61 ms | 37 MB | 3.32 ms | 146 MB | 95.87% |
| | NA2D (Flex) | 1.71 ms | 37 MB | 7.53 ms | 146 MB | 88.58% |
| | HNA (Flex) | 0.61 ms | 37 MB | 3.10 ms | 146 MB | 95.87% |
| | NA2D (NAT) | 0.59 ms | 37 MB | 7.87 ms | 219 MB | - |
| | HNA (NAT) | 0.44 ms | 37 MB | 2.98 ms | 219 MB | - |
| $I = 96 \times 96, W = 11 \times 11$ | SA | OOM | OOM | OOM | OOM | 0% |
| | HSA (FA2) | 0.69 ms | 37 MB | 2.08 ms | 218 MB | - |
| | SA (xFormers) | 6.58 ms | 545 MB | 18.71 ms | 818 MB | - |
| | HSA (xFormers) | 0.81 ms | 37 MB | 2.22 ms | 219 MB | - |
| | SA (Flex) | 1.83 ms | 37 MB | 7.82 ms | 146 MB | 87.89% |
| | HSA (Flex) | 0.61 ms | 37 MB | 3.23 ms | 146 MB | 95.87% |
| | NA2D (Flex) | 1.86 ms | 37 MB | 8.25 ms | 146 MB | 87.50% |
| | HNA (Flex) | 0.61ms | 37 MB | 3.19 ms | 146 MB | 95.87% |
| | NA2D (NAT) | 1.07 ms | 37 MB | 8.38 ms | 219 MB | - |
| | HNA (NAT) | 0.51 ms | 37 MB | 3.06 ms | 219 MB | - |
| $I = 96 \times 96, W = 17 \times 17$ | SA | OOM | OOM | OOM | OOM | 0% |
| | HSA (FA2) | 1.04 ms | 37 MB | 3.21 ms | 218 MB | - |
| | SA (xFormers) | 14.65 ms | 884 MB | 43 ms | 1326 MB | - |
| | HSA (xFormers) | 1.11 ms | 37 MB | 3.36 ms | 219 MB | - |
| | SA (Flex) | 2.79 ms | 37 MB | 11.79 ms | 146 MB | 81.06% |
| | HSA (Flex) | 0.98 ms | 37 MB | 4.50 ms | 146 MB | 93.17% |
| | NA2D (Flex) | 2.87 ms | 37 MB | 12.45 ms | 146 MB | 80.40% |
| | HNA (Flex) | 0.98 ms | 37 MB | 4.50 ms | 146 MB | 93.17% |
| | NA2D (NAT) | 1.22 ms | 37 MB | 9.08 ms | 219 MB | - |
| | HNA (NAT) | 0.77 ms | 37 MB | 4.87 ms | 219 MB | - |
| $I = 128 \times 128, W = 17 \times 17$ | SA | OOM | OOM | OOM | OOM | 0% |
| | HSA (FA2) | 1.86 ms | 66 MB | 5.68 ms | 388 MB | - |
| | SA (xFormers) | OOM | OOM | OOM | OOM | - |
| | HSA (xFormers) | 1.99 ms | 66 MB | 5.91 ms | 390 MB | - |
| | SA (Flex) | 6.12 ms | 66 MB | 23.91 ms | 260 MB | 87.16% |
| | HSA (Flex) | 1.72 ms | 66 MB | 8.15 ms | 260 MB | 96.13% |
| | NA2D (Flex) | 6.41 ms | 66 MB | 26.06 ms | 260 MB | 86.72% |
| | HNA (Flex) | 1.72 ms | 66 MB | 8.17 ms | 260 MB | 96.13% |
| | NA2D (NAT) | 2.18 ms | 66 MB | 16.05 ms | 390 MB | - |
| | HNA (NAT) | 1.38 ms | 66 MB | 8.60 ms | 390 MB | - |

### A.3 THE INFLUENCE OF OTHER PARAMETERS

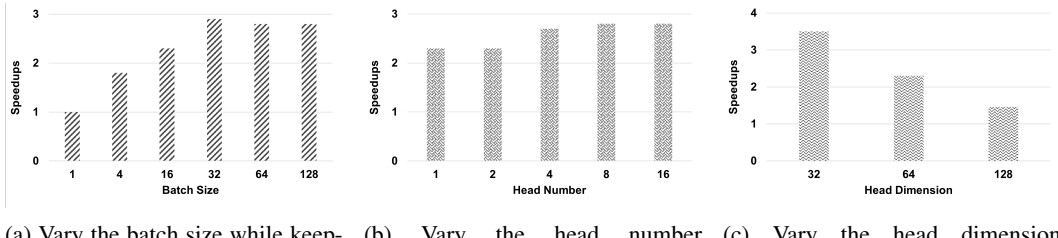

(a) Vary the batch size while keeping head_num=2, head_dim=64, input size =64x64, window size = 8x8 unchanged.

(b) Vary the head number while keeping batch size=16, head_dim=64, input size =64x64, window size = 8x8 unchanged.

(c) Vary the head dimension while keeping batch size=16, head_num=2, input size =64x64, window size = 8x8 unchanged.

Figure 8: Evaluation of speedups by other parameters.

Figure 8 illustrates the impact of batch size, number of heads, and head dimension on speed. These factors do not alter sparsity, but they do impact runtime, and the optimal settings are hardware-dependent. On RTX 3080, when the batch size is 1, WSA and HSA (Flex) run at nearly the same speed. As the batch size increases, the speedup grows, but it stops improving beyond 32, likely due to memory and bandwidth limits. The number of heads has little impact on speedup, whereas increasing the head dimension slows computation because it is more bandwidth-bound.

### A.4 SDPA COMPARISON

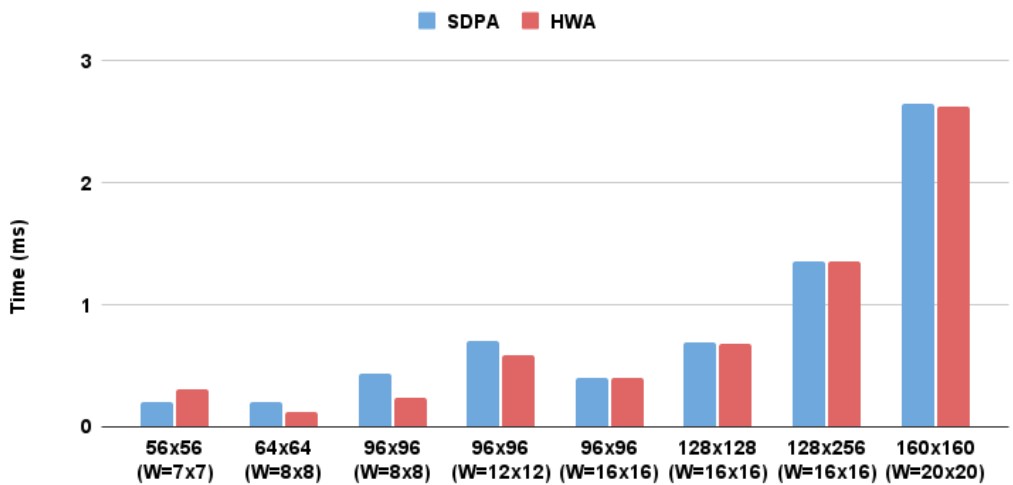

Figure 9: **SDPA vs. HWA (Flex) on RTX 3080.**

Figure 9 compares HWA (Flex) in FlexAttention with SDPA ($F.scaled\_dot\_product\_attention$) across different resolutions and window sizes. SDPA can automatically dispatch to FlashAttention or memory-efficient kernels, accelerating dense attention such as WSA. When sparsity in FlexAttention is high enough, HWA (Flex) remains faster than SDPA. At high resolutions, SDPA benefits from larger windows and can reach speeds comparable to HWA (Flex). For SA/NA, SDPA still materializes intermediates and remains bandwidth and memory-limited, so HSA/HNA perform better.

### A.5 CODE

The code is open-sourced and available at **https://github.com/Yunge6666/Hilbert-Local-Attention**.

## A.6 The Use of Large Language Models

**This paper uses LLM for grammar correction and polishing.**

