# OpenReview forum: "Hilbert-Guided Sparse Local Attention"
_ICLR.cc/2026/Conference — ICLR 2026 Poster_

### Official Review · Reviewer_WpJZ · 2025-11-01

**Soundness:** 2
**Presentation:** 2
**Contribution:** 2
**Rating:** 6
**Confidence:** 2

**Summary:**

This paper proposes a Hilbert-guided sparse local attention method to address the high computational and memory costs of 2D local attention in high-resolution image processing. It reorders image tokens using the Hilbert curve (which preserves spatial locality) to construct contiguous windows/neighborhoods in the 1D sequence, significantly increasing block sparsity. The method achieves substantial speedups for window attention (HWA) and slide attention (HSA) (up to 4× and 18× respectively).  Experiments verify the approach’s practicality.

**Strengths:**

- The Hilbert curve-based token reordering maintains 2D spatial locality while making tokens in windows/neighborhoods contiguous in the 1D sequence. This increases the ratio of empty blocks (reducing partial blocks) and maximizes the efficiency of block-sparse kernels, solving the core bottleneck of traditional row-major ordered local attention.
- Experimental results show significant speedups: HWA outperforms dense window attention by up to 4×, and HSA is 18× faster than conventional slide attention. The method also reduces memory consumption drastically.
- HWT and HNT achieve top-1 accuracy on ImageNet-1K (81.0–81.6%) with only a 0.2% drop compared to baseline models (Swin-T, NAT-mini). This balances efficiency and performance, making it suitable for real-world computer vision tasks.

**Weaknesses:**

This thesis proposes a method for rearranging a sequence of picture patches input to a transformer using Hilbert curves, constructing in a simple way a method that achieves a reduction in the brightness of the attn computation. The paper's experiments are detailed, the narrative is sufficient, and the structure of the lines does not show too many problems. However, from the starting point of the paper, this paper is similar to swin-transformer, both of them carried out technology-based innovation, which is a more skillful design, and I think it is difficult to bring in-depth theoretical insights for the community. But as mentioned, I think this paper is a well-organized and well-structured piece of content, so I think it should be given a weak acceptance.

**Questions:**

see weakness

---

> ### Author Response · Authors · 2025-11-25
>
> **Comment:** We thank the reviewer for the careful review and the positive assessment. As you pointed out, our paper is similar in form to Swin-Transformer in that we also propose a new local attention mechanism. However, our motivation is different: we focus on the mismatch between traditional 2D local attention patterns and modern optimized attention kernels.
>
> ### **Conceptual and system-level insights of Hilbert local attention**
>
> Under the regular row-major sequence layout, 2D local attention becomes highly fragmented in the 1D sequence and at the level of blocks/tiles. This weakens the ability of block-sparse or highly optimized kernels to exploit structured sparsity, even though the attention pattern itself is local. By introducing the Hilbert-curve reorder, we redesign the 1D ordering of tokens within 2D windows and neighborhoods so that tokens from the same window/neighborhood are more contiguous in the sequence and better aligned with the tiling of the underlying kernels. This greatly reduces the number of partial blocks, improves the efficiency of block-sparse kernels, and leads to larger speedups and significant memory savings, while keeping accuracy almost unchanged and without the need to write new kernels.
>
> Therefore, the main contribution of this work lies more on the algorithmic and system side. We highlight “token/sequence layout” as an important research direction that has been relatively overlooked in previous local attention studies but is crucial for efficient implementations. We also provide a concrete, general, and easy-to-use solution that better connects 2D local attention with existing optimized attention kernels. We hope this perspective can inspire future research on efficient vision Transformers and algorithm–system co-design, both in theory and in practice.

---

> > ### Comment · Reviewer_WpJZ · 2025-11-28
> >
> > I agree, I'll keep the rating the same.

---

### Official Review · Reviewer_jRRp · 2025-11-02

**Soundness:** 3
**Presentation:** 3
**Contribution:** 3
**Rating:** 6
**Confidence:** 2

**Summary:**

This work addresses the inefficiency of applying block-sparse kernels to conventional 2D local attention (e.g., window, sliding window), where tokens are non-contiguous in the 1D sequence. The authors propose reordering image tokens along a Hilbert curve before computing attention. This ​​Hilbert-guided ordering​​ increases the ratio of empty blocks that can be skipped by kernels like FlexAttention, significantly accelerating computation. The method is instantiated as Hilbert Window Attention (HWA) and Hilbert Neighborhood Attention (HNA), which are integrated into end-to-end models (HWT, HNT), achieving substantial speedups with minimal accuracy loss on ImageNet.

**Strengths:**

1. Novelty: The idea of optimizing sequence order for block-sparse kernels is creative and addresses a key system-level bottleneck.
2. Generality: The method is model-agnostic and can be plugged into existing architectures (e.g., Swin, NAT) via programmable interfaces like FlexAttention.
3. Strong Empirical Results: Comprehensive experiments show significant speedups (e.g., 4x for HWA, 18x for HSA) and memory savings. End-to-end models validate practicality.

**Weaknesses:**

1. The methods proposed are indeed interesting, but adaptability to non-square inputs or dynamic resolutions is not discussed.
2. Comparisons are mainly against unoptimized baselines. Deeper comparison with highly optimized kernels is needed.

**Questions:**

Regarding the Weaknesses, further discussion and comparison are recommended. However, due to my limited knowledge in this field, the author may selectively reference my suggestions for better presentation.

---

> ### Author Response · Authors · 2025-11-25
>
> **Comment:** We thank the reviewer for the detailed review and valuable comments. We respond to the two points as follows.
>
> ### **Support for non-square inputs and dynamic resolutions**
>
> In practical vision tasks, images or intermediate feature maps usually form a 2D grid of size H×W (H and W do not have to be square and are not fixed). The Hilbert curve can be constructed on any integer grid of size H×W, so our Hilbert reordering is independent of a specific resolution or square inputs. For dynamic resolutions, in the implementation, for each (H, W) size that actually appears when initializing the model, we compute a Hilbert-curve path (from 2D coordinates to 1D positions) and cache it as P_(H,W). When the same resolution (H, W) appears again, we simply reuse the cached path P_(H,W). We rephrased our explanation in Section 3.2, line 259-262.
>
> ### **Comparisons with highly optimized kernels (FlexAttention, NAT, FlashAttention, xFormers)**
>
> In the main body of the original submission, we compared our method with block-sparse kernels such as FlexAttention. As shown in Table 1, directly applying the FlexAttention kernel on WSA can sometimes even be slower than the dense WSA baseline, which clearly reflects the limitation of conventional 2D local attention. In contrast, our HWA, when using the same optimized kernel (FlexAttention), is significantly faster than WSA(Flex). Similarly, the results of SA(Flex) and HSA in Table 3 also show that, under the same optimized kernel, Hilbert local attention is more efficient than conventional 2D local attention.
>
> To address your concern about comparisons with highly optimized kernels, we include additional experiments with FlashAttention and xFormers. Note that these kernels are less flexible than FlexAttention, which supports arbitrary attention patterns.
> FlashAttention natively supports sliding-window (banded) patterns, which match our HSA pattern, so we add HSA results with FlashAttention. However, FlashAttention does not support WSA, SA, or HWA. Therefore, without writing new kernels, WSA, SA, or HWA cannot be implemented on FlashAttention. xFormers does support more attention patterns, allowing us to evaluate WSA, SA, HWA, and HSA using its kernels. The new results have been added to the revised version: Table 1 and Table 3 in Section 4.2, Table 6 in Appendix A.1, and Table 11 in Appendix A.2. In addition, the corresponding discussion has been updated at lines 338–340, 387–391, and 442–450. For convenience, the main numerical results are also shown below.
>
> **WSA vs. HWA with xFormers**
>
> | Input    | Kernel   | Method         | FW Time (ms) | FW Memory (MB) | BW Time (ms) | BW Memory (MB) |
> |----------|----------|----------------|--------------|----------------|--------------|----------------|
> | 56×56    | 7×7      | WSA (xFormers) | 1.21         | 141.3          | 3.15         | 209            |
> |          |          | HWA (xFormers) | 0.26         | 13           | 1.05         | 140            |
> | 64×64    | 8×8      | WSA (xFormers) | 1.74         | 192            | 4.43         | 288            |
> |          |          | HWA (xFormers) | 0.24         | 17           | 1.12         | 163          |
> | 96×96    | 8×8      | WSA (xFormers) | 3.98         | 432            | 9.96         | 648            |
> |          |          | HWA (xFormers) | 0.46         | 37           | 2.16         | 366          |
> | 96×96    | 12×12    | WSA (xFormers) | 9.24         | 612            | 19.76        | 918            |
> |          |          | HWA (xFormers) | 0.78         | 37           | 3.41         | 284          |
> | 96×96    | 16×16    | WSA (xFormers) | 12.17        | 864            | 33.22        | 1296           |
> |          |          | HWA (xFormers) | 0.44         | 37           | 2.40         | 256          |
> | 128×128  | 16×16    | WSA (xFormers) | 24.46        | 1536           | 60.79        | 2304           |
> |          |          | HWA (xFormers) | 0.79         | 66             | 4.74         | 455            |

---

> > ### Author Response · Authors · 2025-11-25
> >
> > **SA vs. HSA with xFormers and FlashAttentionv2 (FA2)**
> >
> > | Input     | Kernel  | Method          | FW Time (ms) | FW Memory (MB) | BW Time (ms) | BW Memory (MB) |
> > |-----------|---------|-----------------|--------------|----------------|--------------|----------------|
> > | 56×56     | 7×7     | SA (xFormers)   | 1.31         | 140.5          | 3.56         | 208          |
> > |           |         | HSA (xFormers)  | 0.28         | 13           | 0.79         | 76           |
> > |           |         | HSA (FA2)       | 0.25         | 13           | 0.75         | 76           |
> > | 64×64     | 9×9     | SA (xFormers)   | 2.47         | 204            | 6.00         | 306            |
> > |           |         | HSA (xFormers)  | 0.37         | 17           | 1.02         | 98           |
> > |           |         | HSA (FA2)       | 0.31         | 17           | 0.95         | 97             |
> > | 96×96     | 9×9     | SA (xFormers)   | 4.79         | 461.9          | 13.49        | 693          |
> > |           |         | HSA (xFormers)  | 0.79         | 37           | 2.22         | 219          |
> > |           |         | HSA (FA2)       | 0.68         | 37           | 2.07         | 218          |
> > | 96×96     | 11×11   | SA (xFormers)   | 6.58         | 545            | 18.71        | 818          |
> > |           |         | HSA (xFormers)  | 0.81         | 37           | 2.22         | 219          |
> > |           |         | HSA (FA2)       | 0.69         | 37           | 2.08         | 218          |
> > | 96×96     | 17×17   | SA (xFormers)   | 14.65        | 884            | 42.50        | 1326           |
> > |           |         | HSA (xFormers)  | 1.11         | 37           | 3.36         | 219          |
> > |           |         | HSA (FA2)       | 1.04         | 37           | 3.21         | 218          |
> > | 128×128   | 17×17   | SA (xFormers)   | OOM          | OOM            | OOM          | OOM            |
> > |           |         | HSA (xFormers)  | 1.99         | 66             | 5.91         | 390            |
> > |           |         | HSA (FA2)       | 1.86         | 66             | 5.68         | 388            |
> >
> > Overall, across different kernels including FlexAttention, NAT, FlashAttention, and xFormers, the Hilbert local attention is consistently more efficient than their corresponding conventional local attention. This shows that the Hilbert local attention is compatible with a wide range of optimized kernels, rather than relying on any single specific implementation.

---

### Official Review · Reviewer_PCkV · 2025-11-03

**Soundness:** 3
**Presentation:** 4
**Contribution:** 4
**Rating:** 6
**Confidence:** 3

**Summary:**

The paper presents a method for improving the computational efficiency of local attention mechanisms in vision Transformers by leveraging the Hilbert curve to achieve higher sparsity in attention computation. By mapping 2D image patches onto a 1D sequence using the spatially-coherent Hilbert traversal, contiguous blocks in the sequence more closely match local neighborhoods, resulting in higher block sparsity when applying block-sparse attention kernels. The paper introduces three Hilbert-guided attention patterns—Window Attention, Neighborhood Attention, and Slide Attention — and demonstrates their integration in the proposed Hilbert Window Transformer (HWT) and Hilbert Neighborhood Transformer (HNT). Comprehensive empirical evaluations show notable speedups with minimal accuracy loss on standard benchmarks.

**Strengths:**

1. This paper provides a systematic and comprehensive framework for different kinds of local attention patterns. The authors identify a persistent gap between theoretical and practical efficiency of sparse/local attention in vision Transformers, especially when block-sparse kernels are applied to row-major sequence orderings.

2. Although the Hilbert curve is not novel in many computational orders in vision models like Mamba, it is exciting to unify the local attention pattern with it.

3.  The paper contains extensive experiments across diverse input sizes, hardware, and key variables, including window/kernel size, block size, attention type, and end-to-end vs. kernel timings. The experiments robustly show that Hilbert-based reordering produces higher empty block ratios, significantly increases throughput, and reduces memory consumption.

**Weaknesses:**

1. The application field proposed in the paper is limited. Since Hilbert local attention has very promising potential in accelerating local attention in vision models, the authors only show results on image classifications. There should be other tasks, including both understanding and generation, e.g., object detection, semantic segmentation, image generation, etc. On these tasks, more SOTA efficient attention mechanisms should also be carefully discussed and compared.

**Questions:**

1. The author-year citation style should be used with brackets.

---

> ### Author Response · Authors · 2025-11-25
>
> **Comment:** We thank the reviewer for the careful review and constructive suggestions. As you pointed out, Hilbert local attention has strong potential for accelerating local attention in vision models, and extending it to broader application scenarios such as detection, segmentation, and image/video generation is an important direction for future work.
>
> ### **Additional classification benchmarks and generality across datasets**
>
> In this submission, results for HWT/HNT are mainly reported on ImageNet-1K, as ImageNet is commonly used to evaluate the representation ability of backbones. On this benchmark, the Hilbert local attention variants achieve accuracy comparable to representative models such as Swin and NAT, while significantly improving the efficiency of local attention. In addition, to verify the applicability of Hilbert local attention under different data scales and image resolutions, supplementary experiments were conducted on CIFAR-10 and CIFAR-100. On these datasets, models with Hilbert local attention achieve accuracy comparable to their corresponding baselines (detailed results are listed below and also provided in Table 5 of the revised version, with the corresponding discussion updated at lines 516–522).
>
> | Model | CIFAR10 (32×32) | CIFAR100 (32×32) |
> |-------|-----------------|------------------|
> | SWIN  | 92.3%           | 76.6%            |
> | HWT   | 92.2%           | 76.4%            |
> | NAT   | 94.2%           | 79.9%            |
> | HNT   | 94.1%           | 79.9%            |
>
> These results indicate that the method does not rely on a specific dataset or a single training configuration and that it has good generality. Taken together, they provide evidence for the feasibility of the proposed method and lay a foundation for transferring it to downstream tasks.
> ### **Extensions to other tasks and efficient attention baselines**
>
> At present, larger-scale HWT/HNT models based on Hilbert local attention are being trained, and there are plans to systematically extend them to object detection, semantic segmentation, and image/video generation using these pretrained backbones. We also acknowledge the reviewer’s suggestion regarding comparisons with more SOTA efficient attention mechanisms on these tasks. In many downstream settings, the head/decoder often accounts for substantial computational cost, so task-specific variants of Hilbert local attention are a promising direction for further study. The code is open-sourced, and updates on larger models and additional downstream tasks will continue to be released in the repository.
>
> ### **Citation format**
>
> The issue regarding the format of citations has been corrected in the revised version.

---

### Meta-Review · Area_Chair_uyJq · 2025-12-28

**Summary:**

Three reviews were received on this paper, which are overall positive.

Reviewer PCkV has concerns on the limited application field (only image classification task is evaluated).

Reviewer jRRp has concerns on the adaptability of the proposed method to non-square inputs or dynamic resolutions, as well as comparison with highly optimized kernels.

Reviewer WpJZ questioned that the paper is a skilful design but lacks in-depth theoretical insights.

**Reviewer Concerns:**

Reviewer PCkV didn’t provide feedback on the authors’ rebuttal. The authors didn’t provide results on tasks other than classification but mentioned they will try more tasks in the future. The ACs think that Reviewer PCkV’s concerns are not fully addressed since the authors didn’t show results on other tasks. Reviewer jRRp didn’t provide feedback on the authors’ rebuttal either. The ACs think that his/her concerns are addressed. Reviewer WpJZ mentioned that his/her concerns are addressed.

**Reviewer Scores:**

While some concerns from the reviewers are not fully addressed, all reviewers think this work is interesting and well-written. The ACs believe they will keep the rating of 6. The final scores are 6, 6 and 6. This paper can be accepted.

---

### Decision · Program_Chairs · 2026-01-26

Accept (Poster)